# Immune Mechanism of Epileptogenesis and Related Therapeutic Strategies

**DOI:** 10.3390/biomedicines10030716

**Published:** 2022-03-19

**Authors:** María José Aguilar-Castillo, Pablo Cabezudo-García, Nicolas Lundahl Ciano-Petersen, Guillermina García-Martin, Marta Marín-Gracia, Guillermo Estivill-Torrús, Pedro Jesús Serrano-Castro

**Affiliations:** 1Epilepsy Unit, Regional University Hospital of Málaga, 29010 Málaga, Spain; maijo.aguilarcastillo@gmail.com (M.J.A.-C.); pablocabezudo@gmail.com (P.C.-G.); nicolundahl@yahoo.es (N.L.C.-P.); guillerminagmartin@gmail.com (G.G.-M.); martamaringracia91@gmail.com (M.M.-G.); guillermo.estivill@ibima.eu (G.E.-T.); 2Biotechnology Service, Regional University Hospital of Málaga, 29010 Málaga, Spain; 3Andalusian Network for Clinical and Translational Research in Neurology (Neuro-RECA), 29010 Málaga, Spain; 4Biomedical Research Institute of Málaga (IBIMA), 29010 Málaga, Spain; 5Neurology Service, Regional University Hospital of Málaga, 29010 Málaga, Spain; 6Department of Medicine, University of Málaga, 29071 Málaga, Spain

**Keywords:** epilepsy, neuroinflammation, neuroimmunology, treatment, epileptogenesis

## Abstract

Immunologic and neuroinflammatory pathways have been found to play a major role in the pathogenesis of many neurological disorders such as epilepsy, proposing the use of novel therapeutic strategies. In the era of personalized medicine and in the face of the exhaustion of anti-seizure therapeutic resources, it is worth looking at the current or future possibilities that neuroimmunomodulator or anti-inflammatory therapy can offer us in the management of patients with epilepsy. For this reason, we performed a narrative review on the recent advances on the basic epileptogenic mechanisms related to the activation of immunity or neuroinflammation with special attention to current and future opportunities for novel treatments in epilepsy. Neuroinflammation can be considered a universal phenomenon and occurs in structural, infectious, post-traumatic, autoimmune, or even genetically based epilepsies. The emerging research developed in recent years has allowed us to identify the main molecular pathways involved in these processes. These molecular pathways could constitute future therapeutic targets for epilepsy. Different drugs current or in development have demonstrated their capacity to inhibit or modulate molecular pathways involved in the immunologic or neuroinflammatory mechanisms described in epilepsy. Some of them should be tested in the future as possible antiepileptic drugs.

## 1. Introduction

The immunologic and/or neuroinflammatory substrate has been found to play a role in the pathogenesis of many neurological disorders. In some conditions, this is considered to be a key element (vg demyelinating diseases of the central nervous system (CNS) [1]), while in others play a secondary role as an adjunct to other mechanisms (vg neurodegenerative diseases [2,3], or other diseases of a genetic or structural basis [4]). The evidence of this transversal involvement of neuro-inflammation and/or immunity in the pathogenic processes of neurological conditions is constantly increasing thanks to major advances in basic research, and epilepsy does not escape from this paradigm.

However, in the case of epilepsy, the classic debate has focused on elucidating whether the underlying immunitary or neuroinflammatory mechanisms are actually the cause or consequence of epileptic seizures [5]. It is incontrovertible that certain inflammatory-based pathological conditions are associated with an increased risk of developing epileptic seizures. Indeed, in most autoimmune diseases, there is a five-fold increased risk of epilepsy in children and a four-fold increased risk in non-elderly adults (aged < 65) [6]. Classic epidemiological studies labeled a large percentage of epilepsies as idiopathic or of unknown origin [7]. However, nowadays, a significant subgroup of these epilepsies is known to have an immune-mediated origin, to the point that the International League Against Epilepsy (ILAE) classification, in its latest update, recognizes immune-mediated epilepsy as one specific etiological category [8].

On the other hand, several studies have shown that epileptic seizures can generate neuroinflammatory activation that, in turn, could be involved in the progression of epileptogenesis [9]. A recent study showed that patients with epilepsy have abnormal levels of plasma inflammatory and neurotrophic markers independent of the underlying etiology, suggesting that such biomarkers may be rather a consequence than a cause of epilepsy [10]. Finally, different models of chemically and electrically induced seizures show upregulation of genes expressed in inflammatory cascades, as seen in patients [11].

Based on this bidirectional relationship between immunity and epilepsy, this review wants to focus on the available evidence of the possibility that biomarkers of inflammation have pathogenic value and, therefore, can become therapeutic targets for epilepsy [12].

## 2. Methods

We carried out a literature search through a Pubmed-Medline (https://pubmed.ncbi.nlm.nih.gov/; last access on 18 February 2022) using as keywords: “Epilepsy”, “Epileptogenesis”, “Brain innate immunity”, “Adaptive immunity”, “Neuroinflammation”, “Therapy” and “Antiepileptogenic drugs”. This search was extended with the bibliographic references found in the selected articles. Following the analysis of these articles, we carry out a narrative review of the state of the matter, including our most recent knowledge related to the following 3 items:Types of immunity and its relationship with the central nervous system (CNS);The mechanisms of immunomediated epileptogenesis;The epileptic disorders related to immunity;The immunomodulatory and anti-neuroinflammatory treatments of epilepsy.

## 3. Results

### 3.1. Types of Immunity and Its Relationship with the Central Nervous System (CNS)

Two main components of the immune system have been characterized, namely the innate (non-specific) immune system and the adaptive (specific) immune system. Their functions and mechanisms are closely linked, and both work synergically against germs or other harmful substances that occasionally may trigger a self-directed immune response against CNS antigens.

The innate immune system provides a general first-line defense in a non-specific manner, mainly through natural killer cells and phagocytes. Their equivalent in the CNS is the microglia [13].

Conversely, the adaptive immune system is considered a second-line defense, driving a targeted immune response mainly through antigen-specific antibodies.

In the last two decades, a better characterization of the role that immunity plays in the different morbid processes that affect CNS disorders has constituted one of the most exciting fields of research in neurology [14,15]. The major focus of research has been conditioned by the discovery of multiple autoantibodies directed against specific CNS antigens that have been proposed as diagnostic markers of these diseases. The target antigens of these antibodies basically include intracellular proteins (cytoplasmic or nuclear enzymes and proteins that act as RNA ligands) and membrane proteins (ion channels and neurotransmitter receptors) [15].

Therefore, it is well known that any immune pathology of the CNS requires the collaboration of a broad spectrum of immune and inflammatory systems [16].

Innate immunity can originate both at the peripheral level (outside the blood-brain barrier (BBB)) or at the central level (central innate immunity). On the other hand, adaptive immunity is always peripheral and requires a disruption of the BBB to exert its effect on the CNS.

#### 3.1.1. Peripheral Immunity (Innate/Adaptive)

The peripheral innate immune response is responsible for the elimination of any challenge in a non-specific manner [17,18]. Its main effectors are neutrophils, monocytes/macrophages, dendritic cells, and natural killer (NK) cells. In addition, native CD4+ T cells are the starting point for the transition toward a specific immune response of the adaptive system, and depending on the inflammatory molecules found in their microenvironment, they can further polarize into various cell lines:T helper cells (Th cells): include three cell subtypes: Th1 cells, defined by the expression of lineage cytokine interferon (IFN)-γ, required for the control of intracellular viruses and bacteria; Th2 cells, defined by the expression of lineage cytokines interleukin (IL)-4/IL-5/IL-13 and the master transcription factor GATA3, which orchestrate the immune reaction against parasites; and Th17 cells, defined by the expression of lineage cytokines IL-17/IL-22, important for the immune response against certain extracellular bacteria and fungi., although they also play a major role in the development of autoimmunity.Regulatory T cells (Treg cells): Its main biological function consists of the suppression of self-reactive cells at the peripheral level.

Differentiation in either cell line is determined by molecular factors produced by antigen-presenting cells and B cells [18,19].

The adaptive immune response is specific to the pathogen presented by antigen-presenting cells, which leads to the clonal expansion of T and B lymphocytes. Each clone that originates from the original T or B lymphocyte has the same antigen receptor as the original and is targeted against the same pathogen or endogenous antigens in case of autoimmunity. In this regard, a field of research has emerged related to the existence of pathogenic autoantibodies against neuronal antigens that cause a broad spectrum of clinical phenotypes by affecting fundamental brain functions [20].

Most of these diseases are the result of an autoimmunization that occurs in the peripheral lymphoid tissue, and one of the most paradigmatic examples is paraneoplastic neurological syndromes (PNS) [21].

Traditionally, CNS has been considered an immuno-privileged organ due to the existence of a cellular barrier that surrounds the brain, mainly composed of endothelial cells, pericytes and astrocytes called the blood-brain barrier (BBB), which constitutes a defensive bastion against both cellular and molecular elements from the bloodstream [22].

For this reason, for all the cellular and molecular elements described in peripheral immunity to act in the CNS, it is necessary that there is a dysfunction of the BBB. This can happen in some circumstances, such as aging or metabolic or infectious insults [23] (Figure 1).

#### 3.1.2. Central Innate Immunity of the Brain

In addition to the peripheral immunity described, an intrinsic immunity originated in the CNS itself has been identified, termed central innate immunity [23], that comprise the following CNS resident immune cells:*Meningeal macrophages*, located in the vicinity of the BBB. They play a main role in immunosurveillance and subsequent presentation of antigens to CD4+ T cells.*Macrophages* near the choroid plexus. Its basic function is also related to immunosurveillance tasks.*Macrophages of the perivascular space*. They are considered a part of the BBB and participate mostly in immunosurveillance and in the recruitment of circulating leukocytes.*Parenchymal cells: Microglia and astrocytes*. They are considered the key elements of central innate immunity but also participate in physiological processes regarding brain development and synaptic plasticity.

Microglia are considered the key activator of this type of immunity. Its function includes the monitoring and detection of disturbances in the brain microenvironment and contributes to restoring homeostasis. In addition to their function as the true “macrophages of the brain”, recent studies have shown that they are also critical for ensuring normal brain development and for tissue repair after a noxious stimulus [24]. Microglia recognizes this aggression and respond developing a phenomenon termed “microglial activation”, a common process in various pathologies including neuropathic pain, neurodegeneration, and traumatic brain injury [25].

The main recognition systems to these noxious stimuli used by the microglia are called innate immune receptors. Among them, the most important are the Toll-like receptor system (TLR), nod-like receptor system (NLR), and inflammasome-associated nucleotide-binding domain, leucine-rich repeat proteins.

All three are membrane-bound molecular chains that are activated after the recognition and binding with peripheral inflammation effectors, which are the “alarm” molecules, fundamentally: *DAMPS (danger-associated molecular pattern)* [26] *as* molecules generated by a heterogeneous group of insults both centrally (traumatic brain injury, cerebral ischemia) and peripherally (radiation, metabolic syndrome, aging [27]); and *PAMPS (pathogens-associated molecular pattern)* [28], small molecular fragments derived from different microorganisms, including HIV or SARS-CoV-2 [29].

After the binding of these innate immune receptors with their peripheral “alarm” ligand, the result will be a subsequent activation of microglia into two states:M1 state or proinflammatory state, which converts microglia into a secretory cell of elements such as cytokines (IL-1β, IL6, TNFα), chemokines (CCL2), and other products such as ROS (reactive oxygen species), NO (nitric oxide), or glutamate, responsible of cell destruction processes of inflammatory origin, also known as pyroptosis [30,31].M2 state or alternative activation, which has the opposite effect, resulting in secretion of anti-inflammatory or neurotrophic factors [32].

Peripheral tissue damage activators can, if they can pass through a dysfunctional BBB, induce conversion to M1 status. The outline of the processes described can be seen in Figure 2.

Astrocytes also exert a key role in central innate immunity under tighter regulation of microglia [33]. It is mainly involved in the control and regulation of immune cells crossing through the BBB.

Under pathological conditions, activated microglia release factors that activate several astrocytic intracellular signaling pathways, such as:Mammalian target of rapamicin (mTOR) pathway;Nuclear factor k-B (NF-κB) pathway;Janus kinase (JAK)-signal transducer;Activator of transcription (STAT) pathway;Mitogen-activated protein kinase (MAPK) pathway [13,34]; andPurinergic signaling.

Figure 3 summarizes schematically some of the main intracellular molecular pathways leading to epileptogenesis.

Subsequently, reactive astrocytes secrete factors that promote changes in BBB permeability, resulting in the recruitment of immune cells into the brain parenchyma, as well as several cytokines, amplifying the initial innate immune response and leading to neuroinflammation. Thus, constant communication (a dialogue) between microglia and astrocytes is a key element in the maintenance of the innate immune response [35].

### 3.2. Mechanisms of Immunomediated Epileptogenesis

#### 3.2.1. Epileptogenesis Mediated by Peripheral Immunity

The epileptogenicity of the molecules arising from the activation of peripheral innate immunity remains a matter of debate [36].

Recent research studies have shown elevated plasmatic levels of these molecules in patients with epilepsy. Particularly, a recent systematic review showed increased levels of IL-1ra, IL-1, IL-6, and CXCL8/IL-8 in several different epilepsy syndromes independently of the underlying etiology [10,37].

In the case of post-traumatic epilepsy, many cytokines are released after severe brain injury for days, accompanied by an activation of ion channels and modifications of receptors associated with neuronal excitation and inhibition. Therefore, microglia and peripheral immunity are activated during hours or days, while plasmatic inflammation persists for weeks and coincides with neuronal loss. Subsequently, mossy fibers sprouting occurs in the hippocampus contributing to increase neuronal excitability [38].

Our understanding of the mechanisms of epileptogenesis related to peripheral adaptive immunity has undergone a radical change thanks to the description and characterization of a new group of autoimmune-based neurological pathologies that has led to the term autoimmune-based epilepsy [15].

Among epileptic disorders related to autoimmunity, it is important to distinguish between immune-mediated encephalitis (especially autoimmune limbic encephalitis (ALE)) and autoimmune-associated epilepsy (AAE), a chronic epileptic syndrome with the capacity for spontaneous recurrence of seizures in the long term [39]. The relationship between ALE and the risk of subsequently developing chronic epilepsy remains unclear and is considered to be mainly related to the nature of the associated neuronal antibody [40]:ALE with Autoantibodies against surface antigens: Infrequently associated with oncological pathology and with suitable response to immunomodulatory treatments [41]. Autoantibodies against surface antigens have an epileptogenic capacity per se, but the risk of developing AAE is lower due to the reversible effect of these antibodies after their removal.ALE with Autoantibodies against intracytoplasmic antigens: Includes classic paraneoplastic syndromes related to onconeural antibodies. Related AAE is characterized by its refractoriness to medical treatments [42,43]. Table 1 summarizes the characteristics of the main paraneoplastic syndromes with antibodies against intracellular antigens.

AAE is increasingly recognized as a clinical entity with its own personality, thanks to the discovery of neural autoantibodies in patients with epilepsies of unknown origin. It is a field in evolution, and the percentage of patients with epilepsy of unknown origin in which an autoantibody can be identified is still modest [56].

Finally, although ALE is often associated with specific anti-neuronal antibodies, it is widely known that those targeting intracellular antigens might be merely biomarkers of the disease, sustained by their inability to reproduce the clinical phenotype in animal models and the prominent T cell-mediated nature of the immune response, which is more commonly associated with the development of AAE following ALE due to neuronal loss and gliosis [26].

#### 3.2.2. Epileptogenesis and Brain Innate Immunity

Recent experimental studies and pathological analyses of brain tissue samples from patients with refractory epilepsy suggest that activation of microglia plays a key role in the etiopathology of epilepsy and seizure disorders [57].

Although the increase in biomarkers of neuroinflammation in patients with epilepsy has been known for a long time, some authors argued that they are a consequence of seizures, while others defended a primary role in epileptogenesis [36].

The biomarkers that have experimental evidence in animal models for modifying neuronal excitability are:**IL-1 Receptor (IL-R1)/Toll-like Receptor (TLR)** [58]: Preclinical investigations in experimental models using pharmacological and genetic tools have identified a significant contribution of interleukin-1 (IL-1) type 1 receptor/Toll-like receptor (IL-1R/TLR) signaling seizure activity. This signal can be activated by ligands associated with infections (PAMPS) or by endogenous molecules, such as proinflammatory cytokines (e.g., IL-1beta) or DAMPS (e.g., high-mobility group box 1 (HMGB1)) [59]. The activation of the IL-1β/IL-1β R axis is strictly linked to the secretion of the intracellular protein MyD88 after the activation of innate immune receptors (especially TLR) during pathogen recognition [31]. This activates an intracellular molecular cascade that, ultimately, can lead to an alteration of neuronal excitability.**TNF-α**: TNF-α affects seizure susceptibility in animal models in a dual pattern. The mechanism of action seems to be exerted through receptors TNFR1 (p55) or TNFR2 (p75) receptor signaling [60]. In general, TNFR1 has been reported to mediate the ictogenic effects of TNF-α, whereas TNFR2 mediates the neuroprotective actions of this cytokine.**High-Mobility Proteins (HMGB1)**: These molecules are components of chromatin and are passively released from necrotic cells and actively released by cells that are exposed to deep stress [61]. Recent studies have described models of epilepsy induced by bicuculline and kainic acid that highlight the nature of HMGB1-TLR4 interactions [62], as well as its role in epileptic recurrence, emphasizing the role of immune-related molecules in epileptogenesis [63].**Cyclooxygenase-2 (COX-2)**: COX-2 is an enzyme synthesizing prostaglandin (PGs) that has also received attention due to its possible involvement in seizure generation [64]. It seems that the presence of COX-2 facilitates the recurrence of seizures in the hippocampus and may upregulate P-Glycoprotein at the BBB, causing AED resistance [65].

In addition to these molecular pathways that have direct evidence, we must not forget that all the astrocytic intracellular signaling pathways outlined in the section dedicated to the basic mechanisms of central immunity are also indirectly related to mechanisms of epileptogenesis.

Finally, both in status epilepticus and in acute symptomatic seizures, there is activation of microglia and astrocytes. Interestingly, experimental evidence found a release of intraparenchymal inflammatory markers such as IL-1β, IL-6, and TNF-α in the hippocampus or neocortex [66].

Table 2 summarizes the main molecular mechanisms related to immune-mediated epileptogenesis at both the peripheral and central immunity levels.

### 3.3. The Epileptic Disorders Related to Immunity

In the previous sections, we have described the main immunological and neuroinflammatory substrates that can be identified in epileptic disorders. It is important to recognize that these substrates have two features:**Universality**. Most epileptic disorders, regardless of their etiology (genetic, structural, autoimmune, infectious, traumatic, etc.), involve a greater or lesser extent the processes described. This universality suggests that these substrates may become valid therapeutic targets regardless of the etiology.**Not compartmentalized**. We will hardly find a pathology that exclusively affects a type of immunity, and a synergic collaboration between the different immune systems is presumed.

Despite these intrinsic features, and mainly considering the therapeutic orientation, it is interesting to establish a classification of epileptic disorders according to the predominant associated type of immunity.

In this section, we intend to establish a classification with a clinical orientation of the main epileptic disorders in which immune mechanisms play a relevant pathogenic role.

#### 3.3.1. Epileptic Disorders Secondary to Systemic Autoimmune Disease

**Primary Antiphospholipid syndrome (APS) or Hughes syndrome**: APS is an immune-mediated disorder characterized by pregnancy morbidity and arterial or venous thrombotic events associated with persistent antiphospholipid antibodies (aPL), including lupus anticoagulant (LA), anticardiolipin (aCL) anti-β2-glycoprotein I antibodies (aB2GPI). Epilepsy has a prevalence of around 6% to 9% in patients with APS [67]. Different mechanisms of epileptogenesis have been proposed in this disease, including aPL-mediated inhibition of GABA receptors and immune-mediated neuronal damage [68].

**Epilepsy-associated Systemic Lupus Erythematosus (SLE)**: SLE is an autoimmune disease in which several organs, tissues, and cells are damaged by adherence to autoantibodies and immune complexes. Anti-DNA antibodies are a subgroup of antinuclear antibodies that can bind to single-stranded DNA, double-stranded DNA, or both and are usually IgM or IgG antibodies. Several recent studies have shown the predominant role of central innate immunity in the pathogenesis of these encephalopathies [69].

**Hashimoto’s encephalitis**: A classical entity included in the steroid-responsive encephalopathy whose clinical expression is an epileptic encephalopathy with focal seizures and frequent secondary generalization, associated with anti-thyroid autoantibodies directed against alpha-enolase, dimethyl-argininase-I, and aldehyde-reductase-I. However, the pathogenic value of these autoantibodies in epileptogenesis is discussed [70].

**Encephalopathy associated with gluten-related disease**: Although the neurological manifestations of celiac disease usually affect other areas of the brain, it is true that people with celiac disease have an elevated risk of chronic focal epilepsy [71]. Neuropathology usually shows predominant T cell infiltration [72] and activation of humoral immunity with the synthesis of antigliadin antibodies.

**Encephalopathy associated with vasculitis and Anti-Neutrophilic Cytoplasm antibodies (ANCA)**: It is a type of vasculitis that rarely produces epileptic seizures in the context of a steroid-responsive encephalopathy [70].

#### 3.3.2. Autoimmune Diseases Primarily Involving CNS

**ALE**: Discussed in the section dedicated to adaptive immunity.

**AAE**: The most frequently detected neuronal antibody has been anti-GAD65, defining a refractory epileptic syndrome with its own personality [73,74]. The neurological anti-GAD65 syndrome is a heterogeneous disease, including stiff person spectrum disorders, cerebellar ataxia, AAE, and ALE in isolation or as part of an overlap syndrome. In general, it is a picture with poor response to immunotherapies, although, in a recent review, epileptic seizures were the manifestation with the best response to long-term immunomodulatory therapy [75].

**Encephalitis with Anti-myelin oligodendrocyte glycoprotein (MOG) antibodies**: Anti-MOG-associated disease is a recently identified autoimmune disorder that occurs in both adults and children as CNS demyelination. However, its clinical presentation includes, in more than 20% of cases, epileptic seizures, which is why it should be mentioned in this section [76].

**Rasmussen encephalitis (RE)**: RE is the prototype of an epileptic disease related to the activation of cerebral innate immunity. Neuropathological studies have provided evidence of a progressive immune-mediated process of neuronal damage with prominent inflammation dominated by T cells, microglial activation, microglial nodules, and astrogliosis [77]. The natural history of RE is well known, with progressive cognitive impairment, brain hemi-atrophy, and continuous partial epilepsy [78].

**ALE due to pharmacological inhibition of immune checkpoints**: The development of oncology immunotherapy has led to the emergence of a new category of neurological immune-related adverse events secondary to the pharmacological blockade of immune checkpoints. The clinical expression is very heterogeneous, but a relevant percentage of these patients develop LE. The pathological substrate of these patients is infiltration of T cells as well as activation of microglia, consistent with a predominance of central innate immunity [79].

In a recent retrospective series that included 63 patients with immune checkpoint inhibitors-related neurologic autoimmunity, 21 (33%) patients whose clinical manifestations included epileptic seizures were described, in most cases, related to LE [80]. A total of 77% of these patients had detectable neural-specific autoantibodies, which implies synergic collaboration between the different immune systems. The drugs most frequently implicated in the literature are **nivolumab** [81,82,83] and **pembrolizumab** [84,85].

#### 3.3.3. Epilepsies Secondary to Structural Etiologies with a Predominant Role of Autoimmunity

**Hippocampal Sclerosis (HS)**: In human epilepsy surgical resections as well as in animal models, involvement of the adaptive immune system was observed in this type of epilepsy. T cell numbers (CD3^+^ as well as CD8^+^) are significantly elevated in HS compared to healthy controls [86]. Secondly, T cell numbers in HS correlated with the degree of neuronal loss.

**mTORpathies (structural pathologies secondary to alteration of the mTOR pathway)**:
**Tuberous Sclerosis Complex (TSC)**: It is a genetic disease with a predisposition to the development of structural alterations of the cerebral cortex called tubers, as well as neoplasms such as subependymal giant cell astrocytoma (SEGA). The pathogenic mechanism is related to central innate immunity abnormalities, mainly mutations leading to a hyperactivation of the mTOR pathway. However, inhibition of this system can be achieved pharmacologically through mTOR inhibitor drugs such as everolimus. This is the first of the genetically based etiologies of which, today, we have a personalized therapeutic strategy that acts directly on the basic mechanisms of epileptogenesis [87].**Focal Cortical Dysplasia (FCD)**: Some FCD produces an activation of cerebral innate immunity, mainly microglial activation. Although their exact pathogenesis remains poorly understood, somatic mutations in the mTOR pathway have been found in FCD type IIb [88].

### 3.4. The Immunomodulatory and/or Anti-Inflammatory Treatments in Epilepsy

All the aforementioned evidence opens the door to the emergence of new antiepileptic therapeutic strategies that surpass the classic paradigm of anti-seizure drugs and inaugurates the field of anti-epileptogenic treatment. Some of them have already been used in the history of epilepsy thanks to the serendipity, such as steroids or adrenocorticotropic hormone (ACTH) for certain catastrophic childhood epilepsies such as West syndrome [89].

Table 3 summarizes the main therapeutic lines according to the degree of development in clinical research.

#### 3.4.1. Established Treatments Used in Clinical Practice

**Corticotherapy**: Steroid therapy has a powerful anti-neuroinflammatory effect due to its genomic effects at transcriptional and post-transcriptional levels, primary exerted on the molecular pathways that converge into the nuclear factor-κβ (NF-κβ) [92], but also prevents microglial activation. It is currently the first-line therapeutic resource in cases of acute onset in the etiologies encompassed within what has been called steroid-responsive encephalopathies (SREAT) such as Hashimoto’s encephalitis [90], paraneoplastic LE [91], lupic encephalopathy, and vasculitis-associated encephalopathy with anti-neutrophilic cytoplasm (ANCA) antibodies [93].

**Immunoglobulins IV (IVIg)/Plasmapheresis** (PLEX): IgIV and PLEX are generally used as a supplement to steroids in the event of acute exposure, but also as maintenance therapy in the event of relapse or chronic conditions. In both types of treatments, the mechanism of action consists in their ability to block or remove pathogenic elements from the circulation (vg autoantibodies or immunocomplexes), but also modulating the proliferation of B cells and plasma cells and production of cytokines [95].

**Immunosuppressors**:**Rituximab**: Anti-CD20 monoclonal antibody used in cases of involvement of adaptive humoral immunity. It is widely used due to its suitable tolerability and safety. Rituximab can be used as both a second-line agent for acute immunosuppression and as a long-term immunosuppressant for recurrent cases. Rituximab, however, does not deplete antibody-secreting cells, which are typically CD20-negative. Therefore, rituximab may work by deleting the antigen-specific memory B-cell populations that secrete the pathogenic antibodies [43].**Azathioprine**: It is an antagonist of the synthesis of purines and, consequently, of the production of DNA/RNA for the proliferation of white blood cells. Azathioprine usually takes 6 to 8 months to be effective, so it is often needed along with the progressive and concomitant reduction in oral steroids [15].**Cyclophosphamide**: It induces cell apoptosis through irreversible alterations of DNA. It is generally reserved for severe cases refractory to other immunotherapies due to the strong immunosuppressive effect and increased risk of adverse events including nausea and vomiting, alopecia, hemorrhagic cystitis, agranulocytosis, infertility, and increased risk of tumors [96].**Mycophenolate mofetil**: This drug depletes guanosine nucleotides preferentially in T and B lymphocytes, and inhibits their proliferation, thereby suppressing cell-mediated immune responses and antibody synthesis [96].**mTOR pathway-modulating drugs**: This line of research has produced the first drugs approved for the treatment of some epileptic disorders of neuroinflammatory basis, such as those included within the tuberous sclerosis complex. **Everolimus** has shown to be effective in the treatment of epilepsy related to tuberous sclerosis with level of evidence I [97]. Additionally, there is an ongoing clinical trial of everolimus in type II FCD (ClinicalTrials.gov Identifier: NCT03198949).

#### 3.4.2. Future Therapeutic Strategies

Novel immunomodulatory strategies will be developed over the next few years due to a better understanding of the basic mechanism underlying the immunopathogenesis of epilepsy. Therefore, most of the following therapies could be proposed.

Among the drugs that modulate innate central immunity, we shall emphasize:**TLR pathway inhibitor drugs**: Research on selective TLR pathway blockade strategy (IL-1β/IL-1βRaxis) has generated some molecules that have shown antiepileptic efficacy in animal models. Among them, perhaps the most promising is **resveratrol** [115].**HMGB1 inhibitor drugs**: Emerging evidence suggest that HMGB1 may contribute to the pathogenesis of epilepsy [62] since **glycyrrhizin**, an HMGB1 inhibitor, exhibits neuroprotective and antiepileptic effects in different animal models of epilepsy [116,118]. However, this drug has not been assessed in clinical trials for epilepsy.**TNF-α inhibitor drugs**: There are four TNF-α inhibitors approved as treatments for ulcerative colitis and/or Crohn’s disease: infliximab, adalimumab, golimumab, and certolizumab pegol [98]. The mechanism of action is based on both the neutralization of TNF-α bioactivity and the induction of apoptosis of TNF-expressing mononuclear cells [99]. **Adalimumab** is a fully human IgG1 monoclonal antibody that specifically binds to TNF-α, which showed to be reduced seizures and functional impairment in patients with RE [100]. Indeed, there is an ongoing clinical trial trying to evaluate the benefit of adalimumab in patients with RE. (ClinicalTrials.gov Identifier: NCT04003922). In addition, infliximab was effective in a patient with relapsing polychondritis and LE [101]. Conversely, golimumab and certolizumab have not been tested for epilepsy.**Drugs modulating T lymphocytes**: A large experience has been gathered about these drugs regarding efficacy and safety due to their use as disease-modifying therapies in multiple sclerosis.**Natalizumab** is a humanized monoclonal anti-α4-integrin antibody approved for the treatment of multiple sclerosis. Integrins are heterodimeric proteins expressed on the cell surface of leukocytes that participate in a wide variety of functions, such as survival, growth, differentiation, migration, inflammatory responses, and tumor invasion, among others [126]. Natalizumab interferes with leukocyte migration across the BBB, which is mediated by interaction between α4-integrin and vascular cell adhesion molecule-1, resulting in a selective CNS immunosuppression due to lower recruitment of immune cells in the cerebral parenchyma [127]. The administration of anti–α4-integrin antibodies showed to reduce seizures in a mouse model of epilepsy [128]. However, a recent phase II clinical trial in refractory epilepsy did not meet the primary endpoint of decreasing seizures, although no adverse events were reported [102]. Further exploration of possible anti-inflammatory therapies for drug-resistant epilepsy is warranted [103]. Interestingly, some cases of RE have been successfully treated with **natalizumab** [104].**Inebilizumab**: this drug is a promising therapeutic monoclonal antibody against the B-cell surface antigen CD19 that has recently been shown to be safe and efficacious in the treatment of neuromyelitis optica spectrum disorder, another antibody-mediated disorder of the CNS [105]. Compared to rituximab, inebilizumab not only depletes CD20+ B cells but also CD20- plasmablasts and plasma cells, resulting in robust and sustained suppression of humoral immunity. There is an ongoing clinical trial phase II with inebilizumab for the treatment of limbic encephalitis with anti-NMDAR antibodies (ClinicalTrials.gov Identifier: NCT04372615).**Cytokine-targeted therapies**: Given the central role of cytokines in many epileptic disorders, these therapies are promising for patients with epilepsy [109]. Among them, we mainly distinguish anti-interleukin drugs (**anakinra and tocilizumab**) and anti-interferon-γ drugs (**situximab and emapalumab**).**Anakinra**, an anti-IL1 receptor, has shown antiepileptic properties in patients with super refractory epileptic status [110], RE [111], and especially in cases of febrile infection-related epilepsy syndrome (FIRES), for which it has been proposed as a first-line therapeutic alternative [112,113]. Additionally, anakinra has been shown to reduce seizures in animal models of anti-NMDAR encephalitis and lithium-pilocarpine-induced epilepsy.**Tocilizumab**, an anti-IL6 receptor, has been reported to be effective in FIRES refractory to anakinra [129]. Currently, there are no research reports with **situximab** or **emapalumab** for epilepsy.**Anti-neonatal Fc receptor (FcRn) therapies**: A novel treatment approach targets the neonatal Fc receptor (FcRn) of several immune cells. The primary function of FcRn is to prevent IgG and albumin from lysosomal degradation through the recycling and transcytosis of IgG, therefore, prolonging its half-life. Antagonism of this receptor causes IgG catabolism, resulting in reduced overall IgG and pathogenic autoantibody levels [106,107]. There is currently an ongoing phase II clinical trial with **rozanolixizumab** (ClinicalTrials.gov Identifier: NCT04875975), a high-affinity human neonatal FC receptor (IgG4P) monoclonal antibody (IgG4P), developed to reduce pathogenic IgG in autoimmune and alloimmune diseases, such as encephalitis with anti-LGI1 antibodies.**Proteasome inhibitors**: Drugs targeting long-lived plasma cells (LLPCs) are a promising treatment for antibody-mediated neurological conditions. Specifically, **bortezomid** is a selective inhibitor of the S26 protasome for which a possible efficacy has been suggested for the treatment of limbic encephalitis with anti-NMDA antibodies. A recent systematic review shows that more than 50% of the patients reported in the literature treated in this way showed improvement. In any case, specifically designed studies are necessary to evaluate this topic [114].**Janus Kinase/Signal Transducer and Activator of Transcription (JAK-STAT) Inhibitors**: Inhibitors of the JAK-STAT pathway are proposed as a future therapeutic target in neuroinflammatory-based processes since they modulate the intracellular signaling pathways of multiple cytokine receptors [120]. Among these molecules, we find **AZD1480, tofacitinib, baricitinib, SAR317461, momelotinib, filgotinib, baricitinib, ruxolitinib, lestaurtinib, or pacritinib** [130], **although lestaurtinib** has any promising studies on animal models of epilepsy [119].**NF-κβ inhibitors**: As mentioned before, steroids preferentially and indirectly act on the NF-κβ transcription complex, although other novel drugs may selectively block this relevant pathway of neuroinflammation. Among them, **dimethyl-fumarate, fingolimod, or teriflunamide** have been widely used in patients with multiple sclerosis [131]. Interestingly, promising results have been reported on animal models of epilepsy treated with **dimethyl-fumarate** and **fingolimod** [122,123]. Emerging experimental findings suggest that **fingolimod** exerts disease-modifying antiepileptic effects based on its anti-neuroinflammatory properties, potent neuroprotection, anti-gliotic effects, myelin protection, reduction in the mTOR signaling pathway, and activation of microglia and astrocytes. Thus, the antiepileptic efficacy of this drug seems to be supported by various mechanisms of action that converge on the modulation of innate brain immunity [117,122].**MAPK pathway inhibitors**: Several drugs perform a selective blockade of this pathway, such as SB203580 [121], macranthoin G, and PD-0325901 (a derivative of CI-1040). The inhibition of p38-MAPK by SB203580 may regulate epileptic activity by decreasing expression levels of adenosine A1 receptor (A1R) and the type 1 equilibrative nucleoside transporter (ENT1) in animal models [121,124].**COX-2 inhibitor drugs**: COX-2 inhibitors have sown anti-seizure properties in various acute and chronic models of epilepsy. **Aspirin, naproxen, rofecoxib, or nimesulide** protected mice from mortality caused by pentylenetetrazol-induced seizure (PTZ) [132].**Purinergic signaling modulating drugs**: Increasing evidence suggests purinergic signaling via extracellularly released ATP as shared pathological mechanisms across numerous brain diseases, including epilepsy. Once released, ATP activates specific purinergic receptors, such as the ionotropic P2X7 receptor (P2X7R). Suggesting the therapeutic potential of drugs targeting the P2X7R for epilepsy, P2X7R expression increases following status epilepticus and during epilepsy, and P2X7R antagonism modulates seizure severity and epilepsy development. JNJ-47965567, a selective P2X7R antagonist, has some evidence in animal models of focal epilepsy [125].

## 4. Conclusions

The avalanche of new evidence revealing the pathogenesis of neuroinflammatory and/or neuroimmunological mechanisms at the origin of many epileptic disorders has broadened the possibility of testing new undiscovered therapies. Several drugs have shown that they can inhibit or modulate the molecular pathways of these immune mechanisms. However, while some therapies such as everolimus have already demonstrated antiepileptic activity in humans, most novel therapies have only suggested their ability to improve seizures in in vitro or animal models, and further studies should assess their potential as antiepileptic drugs.

We find it particularly interesting to point out that these neuroinflammatory mechanisms play a transversal role in the pathogenesis of multiple neurological pathologies. (such as Alzheimer’s disease [2,133] or Parkinson’s disease [134]) and even in psychiatric diseases [135,136,137]. In this sense, a two-way relationship exists between epilepsy and depression and anxiety disorders [138], suggesting a common basis. From this perspective, it is not excluded that the search for new therapeutic strategies that act on the basic mechanisms of epilepsy also influence these comorbidities.

Future studies assessing the efficacy and safety of these therapies in patients with epilepsy are warranted.

## Figures and Tables

**Figure 1 biomedicines-10-00716-f001:**
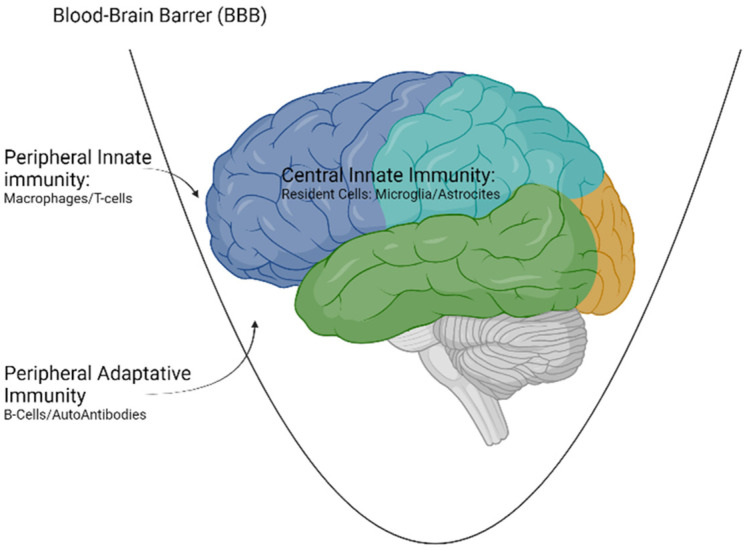
Types of immunity acting at the level of the CNS. Created with www.Biorender.com (last access on 15 March 2022).

**Figure 2 biomedicines-10-00716-f002:**
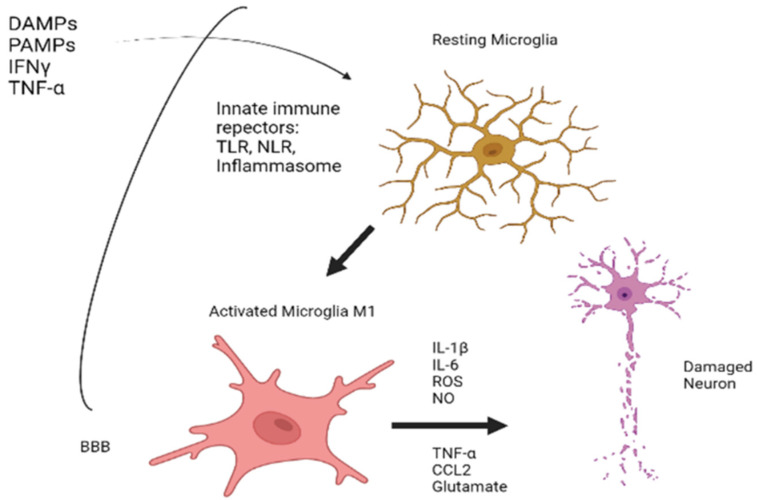
An outline of the processes underlying the activation of microglia. Created with www.Biorender.com (last access on 15 March 2022).

**Figure 3 biomedicines-10-00716-f003:**
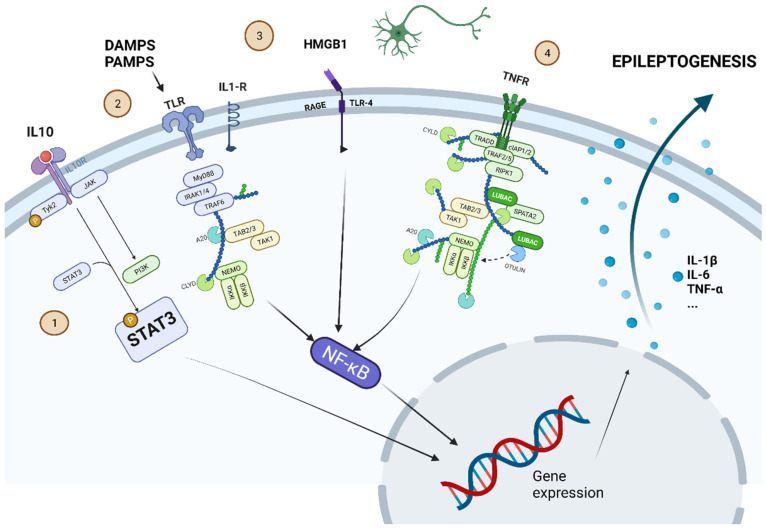
Some of the most important intracellular molecular pathways leading to epileptogenesis are shown. 1. -JAK-STAT pathway: proinflammatory cytokines (such as IL-10) set this pathway in motion. The phosphorylated STAT3 protein can alter gene expression to induce secretion of proinflammatory cytokines (IL-1β, IL-6, or TNF-α). 2. -TLR pathway can be activated by DAMPS or PAMPS. 3. -HMGB1 is produced by cell damage and activates the pathway through its interaction on RAGE-TLR4. 4. -TNFR can be activated by the release of TNF-α. The pathways activated by TLR, RAGE-TLR4, and TNFR converge in the activation of NF-κβ, which also modifies gene expression to induce the secretion of proinflammatory cytokines. Created with www.Biorender.com (last access on 15 March 2022).

**Table 1 biomedicines-10-00716-t001:** Antibodies directed against intracellular antigens.

Antibodies (Epitope)	Clinical Expression	Associated Tumor (>90%)	References
Yo (CDR2L)	Cerebellar ataxia, brain stem encephalitis.	Ovarian carcinoma (>60%), breast carcinoma.	[44,45,46]
Hu (HuD)	Limbic and brain stem encephalitis. Peripheral neuropathy.	Oat cell carcinoma of the lung (>75%). Non-oat cell carcinoma of the lung.	[46,47]
Ri (NOVA1)	Limbic and brain stem encephalitis. Opsoclonus.	Breast carcinoma (>50%). Oat cell carcinoma of the lung.	[46,48]
CV2 (CRMP5)	Encephalomyelitis. Polyneuropathy.	Oat cell carcinoma of the lung (>75%). Thymoma.	[46,49]
Ma1, 2	Limbic and brain stem encephalitis.	Carcinoma of the testicle (50%).	[46,50]
PCA-2 (MAP1B)	Encephalomyelitis. Peripheral neuropathy.	Oat cell carcinoma of the lung. Non-oat cell carcinoma of the lung.	[46,51]
Anti-amphiphysin	Stiff person syndrome. LE.	Oat cell carcinoma of the lung. Breast carcinoma.	[46,52]
SOX-1	Ataxia. Lambert-Eaton syndrome.	Oat cell carcinoma of the lung (>95%).	[46,53]
GFAP	Meningo-encephalomyelitis.	Ovarian teratoma (35%).	[46,54]
Zic4	Cerebellar ataxia.	Oat cell carcinoma of the lung.	[46,55]

**Table 2 biomedicines-10-00716-t002:** Molecular mechanisms underlying immune-mediated epileptogenesis.

Mechanism of Epileptogenesis	Type of Epileptogenesis	Molecular Substrate		References
Peripheral immunity related epileptogenesis	Plasmatic cytokines- mediated epileptogenesis		IL-1ra	[10,36,37]
			IL-1	
			IL-6	
			CXCL8/IL-8	
	Autoantibodies-mediated epileptogenesis	Antibodies against membrane surface antigens	NMDA receptor	[14,39,40,41,42,43,44,45,46,47,48,49,50,51,52,53,54,55]
			GABAa receptor	
			GABAb receptor	
			AMPA receptor	
			Glycine receptor	
		Antibodies against proteins that stabilize potassium channels	LGI1	
			CASPR2	
		Antibodies against enzymes that catalyze the formation of neurotransmitters	GAD65	
		Antibodies directed against intracellular antigens	Yo (CDR2L)	
			Hu (HuD)	
			Ri (NOVA1)	
			CV2 (CRMP5)	
			Ma1, 2	
			PCA-2 (MAP1B)	
			Antiamfifisine	
			SOX-1	
			GFAP	
			Zic-4	
Brain innate immunity related epilleptogenesis	Brain modifying neuronal excitability molecules		IL-1 receptor (IL-R1)/Toll-like receptor (TLR)	[26,32,38]
			TNF-α	
			High-mobility proteins (HMGB1)	
			Cyclooxygenase-2 (COX-2)	
	Astrocytic/microglial intracellular signaling pathways related to epileptogenesis		Mammalian target of rapamicin (mTOR) pathway, nuclear factor k-B (NF-κB) pathway,	
			Janus kinase (JAK)-signal transducer,	
			Activator of transcription (STAT) pathway,	
			Mitogen-activated protein kinase (MAPK) pathway	
			Purinergic signaling.	

**Table 3 biomedicines-10-00716-t003:** Main therapeutic lines according to the degree of development in clinical research. Those drugs that have some study related to epilepsy are marked with an asterisk (*).

Status	Category	Drug	Mechanism of Action	Indications	References
**Established**	**Corticoids**	**Prednisone** **Prednisolone** **Dexamethasone**	Genomic effects at the transcriptional and post-transcriptional level on the molecular pathways that converge on the nuclear factor-κβ (NF-κβ)	**Acute phase of**: **Infantile spasms** **West syndrome** **LE (first line)** **SREAT**	[90,91,92,93,94]
	**Autoantibody removal therapies**	**IgIV** **PLEX**	Remove pathogenic elements from the circulation (vg autoantibodies or immunocomplexes) and elimination of proinflammatory cytokines	**Acute and chronic phases of**: **LE (first line)** **RE**	[94,95]
	**Immunosuppressors**	**Rituximab**	Anti-CD20 monoclonal antibody.	**Acute and chronic phases of**: **LE (second line)**	[43,94]
		**Azathioprine**	Antagonist of the synthesis of purines and production of DNA/RNA for the proliferation of white blood cells	**Chronic phase of**: **LE (second line)** **RE**	[14,94]
		**Cyclophosphamide**	Cellular apoptosis through induction of irreversible DNA alterations	**Acute and chronic phases of**: **LE (second line)**	[94,96]
		**Mycophenolate mofetil**	Inhibits proliferation of T and B lymphocytes, thereby suppressing cell-mediated immune responses and antibody formation.	**Acute and chronic phases of**: **LE (second line)**	[94]
	**mTOR pathway modulating drugs**	**Everolimus**	mTOR pathway modulation	**Chronic phase of**: **TSC** **FCD (ongoing clinical trial; NCT03198949)**	[97]
**Ongoing Clinical trials**	**TNF-α inhibitor drugs**	**Adalimumab *** **Infliximab** **Golimumab Certolizumab pegol**	TNF-α antagonism	**Acute and chronic phases of**: **RE (ongoing clinical trial with adalimumab; NCT04003922)** **LE (retrospective open-label study with infliximab)**	[98,99,100,101]
	**T lymphocytes modulating drugs**	**Natalizumab *** **Inebilizumab ***	Modulation of T lymphocytes	**Chronic phase of**: **Refractory epilepsy (phase II clinical trial with natalizumab; NCT03283371)** **Anti-NMDAR LE (ongoing clinical trial with inebilizumab; NCT04372615)**	[102,103,104,105]
	**Anti-neonatal Fc receptor (FcRn) antibodies**	**Rozanolixizumab ***	IgG catabolism, resulting in reduced overall IgG and pathogenic autoantibody levels	**Acute and chronic phases of**: **Anti-LGI1 LE (ongoing clinical trial; NCT04875975)**	[106,107]
**Open-label Studies**	**Cytokines-targeted therapies**	**Anakinra *** **Tocilizumab *** **Situximab** **Emapalumab**	Modulation of synthesis of cytokines	**Acute phase of**: **RE (retrospective studies with anakinra)** **FIRES (retrospective studies with anakinra and tocilizumab).**	[108,109,110,111,112,113]
	**Proteasome inhibitors**	**Bortezomib ***	Selective inhibitor of the 26S proteasome, preventing the activation of NF-κB	**Acute phase of**: **Anti-NMDAR LE (retrospective studies).**	[114]
**Animal models**	**TLR pathway inhibitor drugs**	**Resveratrol ***	Suppresses NF-κβ induced by TLRs 3 and 4	-	[115]
	**HMGB1 inhibitor drugs**	**Glycyrrhizin ***	Inhibitions of high-mobility proteins	-	[61,116,117,118]
	**Janus kinase/signal transducer and activator of transcription (JAK-STAT) inhibitor drugs**	**Lestaurtinib *** **AZD1480** **Tofacitinib** **Baricitinib** **SAR317461** **Momelotinib** **Filgotinib** **Baricitinib** **Ruxolitinib** **Pacritinib**	JAK-STAT inhibition	-	[119,120,121]
	**NF-κβ inhibitor drugs**	**Dimethyl-Fumarate *** **Fingolimod * Teriflunamide**	Inhibition of NF-κβ pathway	-	[92,117,122,123]
	**MAPK** **pathway inhibitor drugs**	**SB203580 *** **Macranthoin G** **PD-0325901**	Inhibition of p38-MAPK	-	[124]
	**COX-2 inhibitor drugs**	**Aspirin *** **Naproxen *** **Rofecoxib *** **Nimesulide ***	Cox-2 inhibition	-	[64]
	**Purinergic signaling modulation drugs**	**JNJ-47965567 ***	Transient P2X7 receptor antagonism		[125]

## Data Availability

Not applicable.

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
