# Peer review of "Immune Mechanism of Epileptogenesis and Related Therapeutic Strategies"

_biomedicines, 2022, doi:10.3390/biomedicines10030716_

Round 1

Reviewer 1 Report

Comments and Suggestions for Authors
In this review, Aguillar-Castillo and colleagues provide an overview of the heterogeneous mechanism by 
which immunological and inflammatory mediators exert their pathological effects in epilepsy and tried to 
provide a possible therapeutic strategy.
However, this ambitious aim was not always achieved and the review is difficult to read.
Major concerns to address: 
Epilepsy is both a primary disease and a secondary effect of many neurological conditions. Many papers 
show that in pathological conditions characterized by neuroinflammation, there is a higher probability to 
develop epilepsy. 
In this review, the mechanistic link between neuroinflammation and epilepsy should be better described.
Indeed, the bidirectional mechanism of the reciprocal interaction between epilepsy and neuroinflammation 
remains to be understood.
All bulleted lists, in the Results sections, should be replaced by clear tables, with related citations, to make 
the information more immediate and easy to interpret and understand. The titles of all paragraphs should 
summarize the main topic of the paragraph itself, sometimes they are overlapping with each other.
The infiltrating players of the adaptive and innate immune cells are not well described.
The purinergic receptors activated by DAMPs, directly involved in the release of pro-inflammatory 
cytokines, are missing in this review.
How the inflammatory mediators act differentially in acute and chronic phases of epilepsy and how 
immunotherapies could be used with respect to different triggers need to be better explained.
Minor concerns to address:
1.The Title, underlining two aims of the authors: (i) provide an overview of the heterogeneous mechanism 
by which immunological and inflammatory mediators exert their pathological effects in epilepsy and (ii) to 
provide a possible therapeutic strategy, separated by colons, results not really effective
2. It is a review not an article; the abstract must not be divided into paragraphs. As the Introduction 
section, also the abstract must be more direct and concise in the background and in describing the purpose
and novelty of this manuscript. 
3.Material and Methods section must be deleted or changed only in Methods, pointing out the criteria for 
their bibliographic research. I suppose they used PubMed database (https://www.ncbi.nlm.nih.gov/), using 
specific keywords or issues. 
4. Add citations in all presented tables and homogenize the format.
5. Table 3 is never cited in the text.
6. 3.1.1. Peripheral immunity (innate/adaptive) [12], and figure 1 (ref 27) references should be add in the 
text not in the title or in the caption.

Author Response

We thank the Reviewer for his exhaustive work. We have tried to respond to all your comments as accurately as possible. We believe that after these modifications and those suggested by Reviewer 2, the overall quality of the article has improved, and we have managed to make it easier to read.

Point by point responses:

Point 1: In this review, Aguillar-Castillo and colleagues provide an overview of the heterogeneous mechanism by which immunological and inflammatory mediators exert their pathological effects in epilepsy and tried to provide a possible therapeutic strategy.

However, this ambitious aim was not always achieved and the review is difficult to read.

Major concerns to address: 
Epilepsy is both a primary disease and a secondary effect of many neurological conditions. Many papers show that in pathological conditions characterized by neuroinflammation, there is a higher probability to develop epilepsy. 
In this review, the mechanistic link between neuroinflammation and epilepsy should be better described. Indeed, the bidirectional mechanism of the reciprocal interaction between epilepsy and neuroinflammation remains to be understood.

Response: We appreciate this timely comment to the reviewer.

To respond to this comment, we have tried to review the entire text, trying to make it more understandable and easier to read. All changes have been marked in the revision made with the Word editor.

In addition, we have tried to focus the matter better by adding a paragraph in the Introduction explaining the double relationship between epilepsy and neuroinflammation, providing updated references on the studies dedicated to analyzing this problem. 

The part added in the updated version is:

"However, in the case of epilepsy, the classic debate has focused on elucidating whether the neuroinflammatory mechanisms that are usually underlying patients with epilepsy are actually the cause or consequence of the epileptic seizures themselves [5]. It is an incontrovertible fact that the existence of certain inflammatory-based pathological conditions are associated with an increased risk of developing epileptic seizures. Indeed, in most autoimmune diseases, there is a five-fold increased risk of epilepsy in children and a four-fold increased risk in non-elderly adults (aged < 65) [6]. Classic epidemiological studies labeled a large percentage of epilepsies as idiopathic or of unknown origin [7]. However, nowadays a significant subgroup of these epilepsies are known to have an immune-mediated origin, to the point that the International League Against Epilepsy (ILAE) classification, in its latest update, recognizes immune-mediated epilepsy as one specific etiological category [8].

On the other hand, several studies have shown that epileptic seizures can generate neuroinflammatory activation that, in turn, could be involved in the progression of epileptogenesis [9]. A recent study showed that patients with epilepsy have abnormal levels of plasma inflammatory and neurotrophic markers independent of the underlying etiology, suggesting that such biomarkers may be rather consequence than cause of epilepsy [10]. Finally, different models of chemically and electrically induced seizures show upregulation of genes expressed in inflammatory cascades, as seen in patients [11].

Based on this bidirectional relationship between neuroinflammation and epilepsy, this review wants to focus on the possibility that biomarkers of inflammation have pathogenic value and, therefore, can become therapeutic targets against the basic mechanisms of epileptogenesis."

Point 2: All bulleted lists, in the Results sections, should be replaced by clear tables, with related citations, to make the information more immediate and easy to interpret and understand.

Response: Thanks to the reviewer for this suggestion. We have included in the updated version a Table that attempts to summarize all the mechanisms of epileptogenesis described in the text (Table 2), together with the molecular elements involved and the references that support the scientific evidence. The old Table 1 has been included in the new Table 2.

In addition, to make the information more understandable, we have included a new figure (Fig 3), also suggested by Reviewer 2, which shows some of the most important molecular pathways that mediate epileptogenesis in processes dependent on cerebral innate immunity. 

 Point 3: The titles of all paragraphs should summarize the main topic of the paragraph itself, sometimes they are overlapping with each other.  

Response: We agree with this assessment. To make the division of the review clearer, in the material and methods section we have explained that the literature review will be done on four different items:

  1. Types of immunity and its relationship with the Central Nervous System (CNS)
  2. The mechanisms of immunomediated Epileptogenesis,
  3. The epileptic disorders related to immunity
  4. The immunomodulatory and antineuroinflammatory treatments of epilepsy.

The first part aims to describe the basic mechanisms of immunity and neuroinflammation and its relation to CNS from a general point of view.

Parts 2 and 3 try to summarize the evidence on the relationship of these mechanisms with epilepsy from a double point of view: Part 2 analyzes the molecular mechanisms of epileptogenesis related to immunity and Part 3 starts from a clinical perspective to classify neurological disorders with epilepsy in which the basic mechanisms of immunity have an important pathogenic role.

Finally, part 4 tries to summarize the current evidence on the possible therapeutic intervention in the mechanisms described.

We emphasize that the objective of the review is to bring these mechanisms closer to a wide audience that also includes the clinical neurologist, for which Parts 3 and 4, with a more clinical profile, are necessary. Parts 1 and 2 should also be revised depending on who is the intended reader.

Point 4: The purinergic receptors activated by DAMPs, directly involved in the release of pro-inflammatory cytokines, are missing in this review.  

Response: Thank you for alerting us to this significant omission. We have included a section dedicated to this molecular pathway related to epileptogenesis in the different sections and tables, and also, we have included the therapeutic strategies developed using animal models that use this pathway as a therapeutic target.

Point 5: How the inflammatory mediators act differentially in acute and chronic phases of epilepsy and how immunotherapies could be used with respect to different triggers need to be better explained.  

Response: We appreciate this comment to the reviewer. We have included in Table 3 a reference to the moment in which the proposed therapeutic strategy would act (acute phase, chronic phase, or both).

Regarding the moment of action of each of the autoimmune effectors, it has been referred to in the text.

Point 6: Minor concerns to address:
1.The Title, underlining two aims of the authors: (i) provide an overview of the heterogeneous mechanism by which immunological and inflammatory mediators exert their pathological effects in epilepsy and (ii) to 
provide a possible therapeutic strategy, separated by colons, results not effective  

Response: We have changed the title to "Immune mechanism of epileptogenesis and related therapeutic strategies". We believe that this title more accurately reflects the content of the work.

Point 7: 2. It is a review not an article; the abstract must not be divided into paragraphs. As the Introduction section, also the abstract must be more direct and concise in the background and in describing the purpose and novelty of this manuscript.  

Response: OK. We have modified the abstract and the introduction in this sense. 

Point 8. Material and Methods section must be deleted or changed only in Methods, pointing out the criteria for their bibliographic research. I suppose they used PubMed database (https://www.ncbi.nlm.nih.gov/), using specific keywords or issues.   

Response: In accordance with the reviewer's suggestion, we have indicated the keywords used for the bibliographic review. This section has been dedicated to the description of the methods used for the search and the structuring of the selected contents.

Point 9. Add citations in all presented tables and homogenize the format.     Response: We have homogenized the tables and added the citations.   

Point 10. Table 3 is never cited in the text.   6. 3.1.1. Peripheral immunity (innate/adaptive) [12], and figure 1 (ref 27) references should be add in the text not in the title or in the caption.  

Response: We have corrected these incorrections.

Reviewer 2 Report

Dear colleagues, thank you for the interesting review.

I have some concerns/additions.

Figures.

The figures in the manuscript are created in Biorender. With that, according to the Conditions for Publication rights of biorender, you are allowed to publish them in the Journal only if : 1. The ­figure was exported under a paid subscription. 2. Citation of "Created with BioRender.com" appears somewhere in the publication

Please give the information or make the new figures.

Please add the schematic figure of the main routes of therapy used and proposed for the future. I think that such a Figure can help readers.  

Please shorten the parts up to 2. Mechanisms of immunomediated epileptogenesis. The information presented here is a well-known.

I think that the ideas of M Mayes (the inflammatory nature of neurological disorders) are worth discussing, as he was one of the first researchers of that field. Please think about that.

  • Duarte-Silva E, Macedo D, Maes M, Peixoto CA. Novel insights into the mechanisms underlying depression-associated experimental autoimmune encephalomyelitis. Prog Neuropsychopharmacol Biol Psychiatry. 2019 Jul 13;93:1-10. doi: 10.1016/j.pnpbp.2019.03.001. Epub 2019 Mar 5. PMID: 30849414.
  • Morris G, Berk M, Klein H, Walder K, Galecki P, Maes M. Nitrosative Stress, Hypernitrosylation, and Autoimmune Responses to Nitrosylated Proteins: New Pathways in Neuroprogressive Disorders Including Depression and Chronic Fatigue Syndrome. Mol Neurobiol. 2017 Aug;54(6):4271-4291. doi: 10.1007/s12035-016-9975-2. Epub 2016 Jun 23. PMID: 27339878.
  • Maes M, Yirmyia R, Noraberg J, Brene S, Hibbeln J, Perini G, Kubera M, Bob P, Lerer B, Maj M. The inflammatory & neurodegenerative (I&ND) hypothesis of depression: leads for future research and new drug developments in depression. Metab Brain Dis. 2009 Mar;24(1):27-53. doi: 10.1007/s11011-008-9118-1. Epub 2008 Dec 16. PMID: 19085093.

Author Response

Thanks to the reviewer for his exhaustive work. We have tried to adapt the work to your suggestions. All changes are reflected in the review mode of the Word text editor.

Below we answer point by point all your comments.

  Point 1. Dear colleagues, thank you for the interesting review.

Response: We thank the Reviewer for this comment on our work.

I have some concerns/additions.

Point 2. Figures.

The figures in the manuscript are created in Biorender. With that, according to the Conditions for Publication rights of biorender, you are allowed to publish them in the Journal only if : 1. The ­figure was exported under a paid subscription. 2. Citation of "Created with BioRender.com" appears somewhere in the publication

Please give the information or make the new figures.

Response: The three figures in the manuscript have been created by Biorrender, through a paid subscription. This has been noted in the captions of the three figures. We add as an attachment to this comment the invoice that proves our active subscription to Biorrender.com

Point 3. Please add the schematic figure of the main routes of therapy used and proposed for the future. I think that such a Figure can help readers.     Response: Thank you for this important suggestion. We have created a third figure that shows some of the main molecular pathways that mediated epileptogenesis and that are described in the text and may be current or future therapeutic targets for the treatment of epilepsy.   Point 4. Please shorten the parts up to 2. Mechanisms of immunomediated epileptogenesis. The information presented here is a well-known.   Response: Following the reviewer's suggestion, we have removed some paragraphs from the introduction. In any case, given that our objective is a broad public, including neurologists or clinical epileptologists, far from the exhaustive knowledge of neuroinflammatory mechanisms, we think that an introduction that includes at least a brief general explanation of the bases of these mechanisms is necessary. I think that the ideas of M Mayes (the inflammatory nature of neurological disorders) are worth discussing, as he was one of the first researchers of that field. Please think about that.

  • Duarte-Silva E, Macedo D, Maes M, Peixoto CA. Novel insights into the mechanisms underlying depression-associated experimental autoimmune encephalomyelitis. Prog Neuropsychopharmacol Biol Psychiatry. 2019 Jul 13;93:1-10. doi: 10.1016/j.pnpbp.2019.03.001. Epub 2019 Mar 5. PMID: 30849414.
  • Morris G, Berk M, Klein H, Walder K, Galecki P, Maes M. Nitrosative Stress, Hypernitrosylation, and Autoimmune Responses to Nitrosylated Proteins: New Pathways in Neuroprogressive Disorders Including Depression and Chronic Fatigue Syndrome. Mol Neurobiol. 2017 Aug;54(6):4271-4291. doi: 10.1007/s12035-016-9975-2. Epub 2016 Jun 23. PMID: 27339878.
  • Maes M, Yirmyia R, Noraberg J, Brene S, Hibbeln J, Perini G, Kubera M, Bob P, Lerer B, Maj M. The inflammatory & neurodegenerative (I&ND) hypothesis of depression: leads for future research and new drug developments in depression. Metab Brain Dis. 2009 Mar;24(1):27-53. doi: 10.1007/s11011-008-9118-1. Epub 2008 Dec 16. PMID: 19085093.

Response: We sincerely appreciate this comment that has allowed us to make a remarkably interesting translation of the true potential of research in this therapeutic line in epilepsy. Not surprisingly, depression and other psychiatric pathologies have a bidirectional relationship with epilepsy, sharing neuroinflammatory substrates, as shown in the work of the group of Maes et al.

Thus, in the conclusions section we have included the following paragraph that we think gives a new dimension to the work and raises interesting proposals for the future.

The text added is: 

"Of special interest seems to us highlight that these neuroinflammatory mechanisms have a transversal role in the pathogenesis of multiple neurological pathologies, as we pointed out in the Introduction, including Multiple Sclerosis, but also neurodegenerative diseases (such as Alzheimer's disease [2,120] or Parkinson's disease [121]) and even in psychiatric diseases [122–124]. In this sense, the existence of a bidirectional relationship between epilepsy and depression and anxiety disorders is known [125], suggesting a common basis. From this point of view, it is not ruled out that the investigation of new therapeutic strategies that act on the basic mechanisms of epileptogenesis also influences these comorbidities".

We have included the references of the Maes group in the bibliography.

Round 2

Reviewer 1 Report

The revised version of the manuscript has greatly improved.

Minor concerns to address:

  1. Add the refs in table 1
  2. Only moderate English language and style required (i.e. replace we will with appropriate sentence … the authors review the available literature to highlight….)

Author Response

The revised version of the manuscript has greatly improved.

Response: We appreciate the reviewer's comment.

Minor concerns to address:

Point 1: Add the refs in table 1

Response: Required references have been added.

Point 2: Only moderate English language and style required (i.e. replace we will with appropriate sentence … the authors review the available literature to highlight….)

Response: We have carried out an exhaustive spelling and grammar check. We have corrected several minor bugs and changed some sentences to better understand their meaning.

We thank the reviewers again for the splendid work done.